# Man vs. machine: Multi-country experimental evidence on the quality and perceptions of AI-generated research blog content

**Michael Keenan**[1]*, **Naureen Karachiwalla**[1‡], **Jawoo Koo**[1], **Christine Mwangi**[1], **Clemens Breisinger**[2], **MinAh Kim**[3]

**1** International Food Policy Research Institute, Washington, District of Columbia, United States of America, **2** Current affiliation: Consultative Group on International Agricultural Research, Systems Organization, Nairobi, Kenya, **3** Current affiliation: Food and Agricultural Organization, Rome, Italy

‡ With the exception of the first and second authors, the order of the authors was randomized.
* mkeenan@cgiar.org

## Abstract

Academic research is not always available in a form that is accessible or engaging to a non-academic audience, hindering readers' engagement with it. Non-academics, even if highly educated and policy experts in their fields, tend to need research to be presented in a more accessible way than peer-reviewed articles — one example being non-technical blogs. However, writing these requires some effort from researchers. Artificial Intelligence (AI) tools can make academic research easier to understand by summarizing and simplifying academic papers much more quickly than researchers can, making it easier for researchers to produce such summaries. However, disclosure of AI use may lower readers' perceived quality of and trust in the blog, generating a trade-off for the researcher. In this paper, we evaluate an 11-country experiment cross-randomizing a blog's actual and reported author as AI or human. We find that research stakeholders rate the quality of AI-generated blogs marginally lower than human-written ones (p < 0.1), but disclosure of AI use offsets the negative effect (p < 0.1). The study sample consists of policy-relevant stakeholders who typically engage with academic research; they are highly educated and include thematic specialists. Indeed, findings indicate that this audience interprets "accessibility" differently, preferring slightly more technical summaries of research. The nature of the respondents may thus explain the particular findings in this study, suggesting that researchers should tailor their prompts for their intended audience. There are no effects on readers' reported likelihood of engaging with the blog or on beliefs about others predicted engagement with it. Consequently, we hypothesize that researchers can leverage AI to communicate their research more easily without a penalty from disclosing its use.

**Data availability statement:** We have made the data available on the IFPRI Dataverse at: https://doi.org/10.7910/DVN/GETDGX.

**Funding:** This work was supported by the CGIAR Initiatives on National Policies and Strategies (NPS) and Digital Innovation (DI) and the Bill & Melinda Gates Foundation through the Generative AI for Agriculture project (INV- 047346). The funders had no role in study design, data collection and analysis, decision to publish, or preparation of the manuscript.

**Competing interests:** The authors have declared that no competing interests exist.

## Introduction

Academic research is only valuable to the extent that it is understood, appreciated, and used by the audiences for whom it is intended. For many researchers, the ultimate goal is for readers to consider their findings and engage with them – for example, by using findings in their own research or in practical decision-making. Many non-academic readers tend not to read full academic papers; rather, shorter, non-technical summaries such as blogs are often preferred [1]. Generative Artificial Intelligence (henceforth, AI) that uses Large Language Models (LLMs) can facilitate such communication by easily creating simpler communications and making research more accessible [2]. However, there may be biases in perceptions of AI as well as ethical concerns.

AI has been shown to reduce knowledge workers' time on various tasks by 18% on average, improve expert-assessed writing quality by 40% [3], and increase the efficiency of programming by between 26% [4] and 56% [5]. Globally, 58% of people use AI "on a regular or semi-regular basis," but in low-income countries that figure is 80% [6]. However, even with recent improvements, trust in AI remains a concern, with only 39% of people globally trusting AI-generated information since AI can make errors, be biased [7], spread misinformation [8], or make up information [9–11]. Indeed, the broader scientific community has been grappling with ensuring the responsible use and disclosure of AI in research [12–14], including the peer-review process [15].

In this paper, we examine how AI might serve researchers in more easily communicating their work to various consumers of research (such as other researchers and academics, policymakers, and other non-technical stakeholders). The seminal paper by Akerlof [16] and a more recent paper specifically related to communication by Gans [17], provide useful frameworks by which to think about such communication and the introduction and perceptions of AI. Readers incur time and effort costs of consuming information but derive utility from high-value communications. Writers (researchers) decide whether to communicate their findings via simpler summaries (blogs) by balancing the utility they derive from reader engagement against the effort costs of producing accessible and compelling content. Accessibility, which means using less jargon, engaging titles, and polished visuals, lowers the cognitive effort cost required for readers to engage (for example, by looking up the underlying research or sharing it with others) but imposes additional effort costs on researchers. Readers, in turn, base their engagement decisions on the perceived value of a blog, which they infer from observable signals of researcher effort in making the blog accessible. Higher perceived effort is often equated with higher value research as it is less valuable for authors of low-value research to invest effort in making their work more accessible. Consequently, these signals help address the inherent information asymmetry between researchers who know the true value of their research and their effort level, and readers, who can only estimate it.

The use of AI disrupts this traditional signaling dynamic by significantly reducing the costs of producing accessible and polished content. While AI lowers effort for both researchers and readers, it also undermines the reliability of traditional effort-based

signals as it introduces ambiguity about effort. Researchers can use AI to create content that appears high-effort even when little effort has been invested, exacerbating information asymmetries and making it harder for readers to assess the true value of the research. Most people are ambiguity-averse [18]; in such cases, readers may assume average quality, leading to reduced engagement, a dynamic akin to the "Market for Lemons" [16,17]. Researchers must also decide whether to disclose the use of AI, as transparency may mitigate signaling ambiguity but could simultaneously signal low effort, thereby reducing perceived value. The return to using AI and of disclosing its use is thus an empirical question.

In a pre-registered experiment, we implemented a four-arm factorial design with 366 respondents who are relatively highly educated and connected with policy-relevant research. They were randomly assigned to read either a human-written or AI-generated blog (testing accessibility) and were randomly informed that the blog was either written by a human or generated using AI (testing signaling). We test whether AI-generated research communication affects perceived quality and whether clear disclosure of authorship mitigates potential ambiguity. An online survey was distributed across 11 countries and respondents evaluated the blogs based on accessibility-related characteristics and reported their intended engagement with the research, as well as their perceptions of how others might engage with it.

We find that, while the AI-generated blogs are objectively more accessible – they are written at a lower grade level, as measured using the Flesch-Kinkaid score [19] – when respondents read an AI-generated blog, they rated it lower in quality, on average, though the magnitude of the effect is small. Weak signaling may have introduced ambiguity about the author, reducing perceived quality of the blog and thus the research. However, this negative effect is almost completely offset if respondents are told that the AI-generated blog was in fact AI-generated, indicating that a disclosure statement (a truthful signal) can successfully offset the negative effects of ambiguity.

In exploratory analysis, we examine potential mechanisms. We do not find differences between experimental arms by the grade level of the text, so it is not the case that the AI-generated blogs were of lower quality. Instead, there may be a bias in perceptions about AI. Additionally, more than 80% of respondents in our sample have master's degrees and above, potentially indicating that accessibility may not have the same value to this audience – they may prefer more technical writing and may be more familiar with AI and with disclosure statements. Researchers should thus tailor the use of AI to their audience. While ours may not be a representative sample of all consumers of research, arguably, this is a relevant sample that includes academics, policy-makers, donors, and practitioners associated with a well-known policy research institute with a wide reach.

Women and those who report having previously used AI also rate AI-generated blogs negatively, but truthful disclosure of AI use offsets the negative effect more for women and those who have used AI. One possibility is that those who have not previously used AI face higher ambiguity and thus the disclosure statement may reduce ambiguity more for them. Those who have used AI may have guessed that researchers employed AI in writing the blog; [20] notes that some people can detect use of AI as it tends to have a distinct writing style, which would imply a lower level of ambiguity in authorship for this group. There are no differential effects by those who report trusting AI to write high-quality blogs; another dimension across which behavioural aspects play a role. Note that statistical power is lower in these interacted specifications compared to the primary specifications, so we interpret these results with some degree of caution. On respondents' intended engagement and their beliefs about others' engagement with the blog, in all four experimental arms, we cannot reject that the difference in the likelihood of engagement across groups is zero.

The findings suggest that, at least in this sample, the respondents valued the elimination of ambiguity in the signal via the disclosure statement and did not penalize the potentially perceived lower effort in generating the blogs using AI. We conclude that, provided that disclosure statements are included, researchers may reduce their cost of effort by using AI for research communication without compromising the perceived value of the communication or respondents' engagement with it.

This paper contributes to a well-established literature on communication and information frictions, focusing on asymmetric information. The problem has been studied in the context of agricultural technology adoption and medicines

when they may be counterfeit (sellers know the quality of the product but buyers do not) [21–25] and in advertising where companies want to signal the value of their products [26], among others. In all cases, producers cannot credibly signal or communicate the value of their products via prices – just as researchers cannot perfectly signal the value of their research via their effort. The research broadly finds that "labeling" products with third party certifications, providing warranties, product assurance schemes, minimum quality standards, or information aggregation through reviews, can create credible signals of quality [27–32]. We build on this literature by studying a relatively recent but increasingly used method in communication, AI, and the potential of reducing information asymmetries that are naturally associated with it. A simple solution exists, akin to certification: AI disclosure statements can reduce information asymmetries that hinder communication since they provide a truthful signal about authorship, allowing readers to more precisely judge the underlying value of a communication. Ours is one of the first experiments to causally study the effects of AI on perceptions of quality and engagement for an important and widely used but potentially tedious task in academia, and we do so in a real-world setting.

We also contribute to the burgeoning literature on the effects of AI on knowledge work. Recent evidence has centered around the role of AI in affecting knowledge workers' jobs and productivity [3,33]. While [33] and [34] report complementarities between human and computer generated work, estimating efficiency gains of 27% and 60%, respectively, [35] does not find evidence of these complementarities. Despite this, there remains scant causal evidence on the effects of AI on very high level knowledge work, such as that of academics and other researchers. One paper by [15] examines the use of AI in peer-review. Another by [36] studies the acceptability of AI use in Universities, [37] explore instructors' perceptions about AI in Palestinian universities, [38] studies techno-stress to professors in Palestinian universities, and [2] and [8] note that idea generation, feedback, coding, and summarizing and improving text are some of the main use-cases of AI for economic research and that those functions are constantly improving, rendering this paper quite consequential.

Finally, while the literature has touched on opinions about AI writing and disclosure in academic writing [13], research ethics [39], summarizing literature and the potential hallucinating that may occur [40,41], there is very little causal evidence on the perceived quality of AI-generated research outputs, and none on how users may interact and engage with AI-assisted academic research outputs. Our paper is most closely related to [42], who conduct an incentivized experiment that varies both the actual and reported authorship of a paragraph, asking participants to evaluate their trust in its content. Our findings complement theirs, demonstrating that participants do not distrust AI-generated text when the author's identity is disclosed. We build on their work by employing an experiment in a real-world context across a sample of eleven countries and by additionally exploring participants' reported likelihood of future engagement, and beliefs about others' potential future engagement with the content. Given the importance of communication in academic research and the increasing role of AI in knowledge work, studying how the two interact to potentially affect real-world outcomes is a key contribution.

## Conceptual Framework

We draw on the general theory and frameworks first elucidated by Akerlof [16] and adapted for communication by Gans [17], as well as insights from the broader literature, to examine how readers perceive the quality of research communication and how this perception may influence future engagement. The implied framework of Wiles et. al. [43] is also strongly aligned with Gans' framework. The framework focuses on two key decisions: (1) researchers decide whether to communicate their findings through an accessible communication – a blog, and (2) readers decide whether to engage with the blog – read it, share it, look up the original paper, and others. Readers are more likely to engage with blogs that are written in an accessible manner, meaning minimal jargon, engaging titles, polished figures, and clear explanations. Researchers derive utility from having their research absorbed and utilized by readers, and their decision to write a blog is influenced by the likelihood of reader engagement. However, producing a high-quality, accessible blog entails effort costs, which serve as a signal to readers regarding the value of the underlying research.

 

The value of research varies, with some studies being inherently more valuable (high-value research, H) than others (low-value research, L). Ex-ante, readers lack perfect information about the value of the research and instead rely on signals such as the effort invested in making the research accessible to infer its quality. Notably, lower-quality research is more difficult to translate into a high-quality, accessible blog, so at some threshold, it is too costly for researchers producing L research to make their research more accessible. Consequently, researcher effort emerges as a key signal for communicating research's potential value. While other factors, such as academic credentials, can also serve as signals, these are more closely related to credibility than effort [44,45], and we hold them constant across treatments by standardizing the wording about authors. Accordingly, the greater the perceived effort exerted by the researcher, the stronger the signal, leading to a higher perceived value and an increased likelihood of reader engagement. However, an inherent information asymmetry exists: researchers are aware of their true effort costs and the value of their research, whereas readers are not.

Readers, in turn, derive utility from clear and valuable information, basing their engagement decisions on the expected value of the content. However, engaging with research communication also imposes effort costs, such as cognitive effort and time investment. Given limited attention and cognitive resources, readers prefer accessible (concise and easily digestible) content, which reduces their cost of engagement. Researchers, therefore, face a trade-off: investing in accessibility increases their own effort costs while simultaneously lowering the effort costs for readers that would increase engagement with their research if they put in the effort.

The introduction of AI disrupts this traditional signaling mechanism for both researchers and readers (a mechanism also noted in [46] and [47]). AI can simultaneously reduce effort costs for both parties by streamlining the process of making research more accessible. However, it also exacerbates information asymmetries, particularly concerning the signaling of researcher effort across both H- and L-type research. By leveraging AI, *researchers can artificially present low-effort work as high-effort work*, thereby weakening the reliability of effort as a signal. When signals become ambiguous, readers face greater difficulty in discerning research quality, leading them to assume an average quality level and subsequently reducing their engagement with research communication.

Two trade-offs emerge: First, a researcher will opt to use AI if the benefit of doing so – including the potential increase in reader engagement – is higher than the cost of writing the blog themselves. Additionally, researchers must choose whether to disclose AI assistance in blog creation, which would eliminate ambiguity. Disclosure is optimal only if the benefits of reducing ambiguity outweigh the potential costs associated with revealing lower effort. These trade-offs form the central questions of our analysis.

## Methods

### Intervention design

Our experiment randomly varies signaling and accessibility to assess how research consumers perceive blog quality and their likelihood of future engagement. We conduct an online experiment leveraging IFPRI country office mailing lists, with two cross-randomized interventions: participants read a blog either written by a researcher or generated by ChatGPT-4 (AI) and were told – accurately or inaccurately – whether it was human- or AI-authored. Note this approach is not uncommon; see [35,48], and [42]. Participants were later told the actual authorship of the blog. We received ethical approval from IFPRI's IRB (IRB number DSG-23–0833). Informed consent was obtained through the online survey – respondents were provided with a consent statement and were able to proceed with the survey or exit the survey at that moment or at any other time.

We used academic papers and their existing corresponding human-written blogs, written by IFPRI researchers, focusing on food systems since this is most often the topic stakeholders receive information about from IFPRI. AI-generated blogs were created using detailed uniform prompts in ChatGPT-4 (see Appendix C for the full prompt). Prompts were input using a piecemeal, iterative process – the researcher first prompted GPT-4 to create a title, then an introduction and research overview, key findings, policy recommendations, and a conclusion. The prompts were derived from best practices in prompt engineering [49,50].

A research assistant inserted key figures from the source paper, just as researchers would select them even if they used AI to help generate the blog [35,51]. They also checked for glaring errors (none were found). AI content creators and research assistants did not see the human-written blogs. AI blogs matched human blogs in length (within 60 words, < 10%) and covered topics relevant to each country (per IFPRI country directors' advice), within the topic of food systems, to ensure that highly divergent topics would not substantially affect the results, on average. Blog format was standardized, with no branding or IFPRI affiliation. All papers and human-written blogs were published before this experiment was conceived and were authored by academics rather than students, ensuring no bias toward human- or AI-generated content.

Before presenting the blog, we provided a disclosure statement to a randomly selected half of the respondents based on Elsevier journal regulations (Appendix B). The four treatment arms are (actual author is the first word and reported author is the second):

- AI-Human and AI-AI: "During the preparation of this blog, a team of local and international researchers used Artificial Intelligence to generate the content and structure of the blog. The authors have reviewed and edited the content as needed and take full responsibility for the content of the blog."

- Human-Human and Human-AI: "This blog was written by a team of local and international researchers."

See also Table 1. The AI-AI and Human-Human arms test accessibility by comparing perceived blog quality based on factors like catchy titles and reduced jargon, while AI-Human and Human-AI test signaling by examining how author disclosure affects perceived effort. The models predict that if AI-generated blogs are more accessible, they should receive higher quality ratings. However, while disclosing AI authorship removes ambiguity, it may also signal lower effort, making the overall effect difficult to predict ex-ante.

## Sample and data

An online survey with 366 participants was conducted leveraging IFPRI mailing lists and social media accounts across eleven countries with IFPRI offices. The countries included Bangladesh, Egypt, Ethiopia, Ghana, India, Kenya, Malawi, Nigeria, Rwanda, Sudan, and Uganda, with the largest number of participants coming from Bangladesh (89), India (59), and Kenya (52). Participants were assigned to treatment groups within the online survey that used the survey software SurveyCTO. The method used was systematic, stratified randomization. Respondents first provided demographic and employment details (gender, age, education, position, and organization type) and were dynamically randomized, within country, on six strata: gender (male/female), and position (junior, mid-level, senior-level). The first respondent within each stratum was randomly assigned to one of the four treatment groups, the second to one of the three remaining, and so forth.

The study was conducted in four waves: October 2023, November 2023, January 2024, and March 2024. Randomization was not conducted within each wave; instead any leftover stratum was carried over into the subsequent wave. In the first and fourth waves, heads of the country offices emailed their mailing lists with study details and survey links, followed by a reminder two weeks later. The second and third waves used social media recruitment via IFPRI's LinkedIn and X accounts. To incentivize participation, respondents were informed that ten randomly selected respondents would win 100 USD each. Winners have since been paid. While the original plan was to select winners based on the accuracy of

**Table 1. Experiment design.**

| | | Actual Author | |
| --- | --- | --- | --- |
| | | AI-generated | Human-written |
| Reported Author | Human-written | AI-Human | Human-Human |
| | AI-generated | AI-AI | Human-AI |

respondents' predictions, specifically by comparing their beliefs about how others would engage with the blog to their own self-reported intended engagement, country office directors preferred random selection to simplify the logistics. This led to a slight deviation from the pre-analysis plan.

## Measures

We assess respondents' perceptions of blog quality, intended future engagement, and their beliefs about others' engagement. Our four primary outcomes are: (1) the unweighted average of 10 quality attributes, (2) a weighted average of 9 attributes based on respondents' preferences for blogs, (3) the unweighted average of five statements on intended engagement, and (4) the unweighted average of five statements on beliefs about others' engagement. Robustness checks use alternative index calculation methods.

Before reading the blog, respondents rated the importance of seven blog characteristics related to accessibility documented as being important to the quality of a communication and valued aspects of academic research communication [45,52–54]. They included the catchiness of the title, amount of detail, clarity of the rationale, visual presentation, length, ease of understanding, and the clarity, relevance, and detail of the policy recommendations. These preferences were measured on a 0–4 Likert scale (0 = 'not important at all', 4 = 'very important').

After reading the blog, respondents rated its quality on 10 attributes, using a Likert scale ranging from 0 = 'strongly disagree' to 4 = 'strongly agree' that the blog is high quality in that attribute. They included the seven characteristics listed above plus an additional measure of whether the blog had an appropriate tone. Due to a coding error, tone was not exported from the survey in the preference questions, but was in the quality ratings. Additionally, policy recommendations were evaluated as one item in preferences (clarity, relevance, and detail) but split into three in the questions about quality ratings.

Five quality measures were calculated: (1) the unweighted average of the 10 attributes, (2) a weighted average based on respondents' stated preferences, where the weights reflect each attribute's relative contribution to total preference ratings, with the three components of policy recommendations equally weighted, (3) an unweighted average across the 9 attributes for robustness, (4) an index calculated from a Polychoric principal component analysis, and (5) a binary indicator for whether respondents rated the blog's overall quality as 'high' or 'very high' from a separate question (the latter two also being robustness checks).

Engagement and beliefs about others' engagement were measured using five Likert-scale statements (0 = 'very unlikely', 4 = 'very likely') on whether they or others would share the blog, re-read it, look up cited studies, explore related studies, or contact the authors. Three own and others' engagement indices were created: an unweighted average, the first principal component of a Polychoric principal components analysis, and the total of 'likely' and 'very likely' responses. We also recorded the log of reading time in minutes and report it as a measure of own engagement.

Finally, we calculated blog readability using the Flesch-Kincaid Grade Level formula [19] as an objective measure of accessibility. The formula estimates the U.S. school grade level required to understand a piece of text, calculated based on the average number of syllables per word, words per sentence, and sentences per word, with higher values indicating more complex text suitable for higher-grade readers. A limitation of this metric is that it was developed for American English and U.S. school grades; thus, it may contain measurement error when applied to this context.

## Analytical and statistical methods

We estimate an OLS model (using Stata 18) as follows to test effects of the treatments with regards to accessibility, signaling, and disclosure:

$$Y_i = \beta_0 + \beta_1 AI_{generated,i} + \beta_2 AI_{reported,i} + \beta_3 AI_{generated,i} * AI_{reported,i} + \beta_4 X_i + \delta_j + \varepsilon_i \tag{1}$$

where $Y_i$ is the outcome of interest, $AI_{generated,i}$ is an indicator variable equal to one if the blog was generated using AI, and $AI_{reported,i}$ is an indicator variable equal to one if the blog was reported as being AI-generated. $X_i$ is a vector of demographic controls (see Table S1 in S1 File for the list of variables included). Relative to a human-generated blog reported as human-authored, $\beta_1$ identifies the effect of an AI-generated blog (accessibility), $\beta_2$ identifies the effect of respondents being told the blog was generated by AI when it was not (signaling), and $\beta_3$ identifies the differential effect of an AI-generated blog reported as AI-generated versus reported as human-written (disclosure). We include strata (country, gender, and seniority) and wave fixed effects $\delta_j$ and heteroskedasticity-robust standard errors (clustering is at the individual level, the unit at which treatment was assigned).

We report conventional standard errors as well as randomized inference (RI) *p*-values in squared brackets. When discussing statistical significance, we always refer to the RI *p*-values. Our preferred specification includes demographic control variables due to our small sample size, but we report specifications without controls as well. There are four reasons that we use OLS for all of our outcomes despite the sums of engagement behaviours being ordinal. First, our primary interest is to compare differences in means rather than the probability of being in a particular category. Second, when there are 2–7 categories, OLS often fares just as well as ordered logit [55,56]. Third, in specifications with interactions (in our case, there are triple interactions), the interpretation of the odds ratios from ordered logit is extremely difficult. Fourth, we preferred to keep the specifications consistent over the outcomes to ensure comparability.

To examine mechanisms, we conduct heterogeneity analysis across several dimensions by including the variable itself and interacting it with each of the three variables in Equation 1. For these RI *p*-values, we permute each variable in the interaction separately and report both (or all three for triple interactions). Comparing the interactions between that dimension and each of "AI-generated" and "AI-reported" allows us to measure the difference between those with and without that characteristic.

We assess balance in demographic characteristics across the four experimental arms, and, across 18 variables and 108 tests (Appendix Table S1 in S1 File), we find only seven statistically significant coefficients, indicating that the experiment was balanced.

## Results

### Characteristics of the sample

Our sample is highly educated and specialized, an important point to note for interpretation. The average respondent is 42 years old, with just under a third female and over 80% holding a master's degree or higher. Half are senior-level professionals, while the rest are evenly split between junior and mid-level roles. About 40% work in research institutes, 25% in NGOs, and 15% in the private sector. The most common fields are human development (60%), agriculture (60%), and the environment (25%) (as this was a multiple-select question, the totals do not add up to 100). Additionally, 35% report having high potential to influence policy. These characteristics suggest our sample is well-positioned to engage with research and at least inform, if not influence, policy decisions.

The most valued blog attributes were clear recommendations (mean 3.66 out of 4) and easy-to-understand text (3.61 out of 4) (see Appendix Table S2 in S1 File). Blog quality perceptions were high, with an average rating of 3.4 (out of 4), and 60% of respondents rating them as 'high' or 'very high' in quality. Respondents preferred that blogs be 7.8 pages with 6.5 figures — much longer than the actual blogs. Engagement levels were moderate (2.1–2.8 out of 4), with predicted engagement by others averaging half a point higher. Additionally, 60% of respondents had previously used AI, mainly for shortening text or correcting grammar. For context, 80% of people in low-income countries use AI tools on a regular or semi-regular basis, compared to 58% in high-income countries. Thus, our sample's use of AI is somewhat lower than that of the general population in low-income countries [57].

## Perceptions of Quality

Fig 1 and Appendix Tables S3 and S4 in S1 File (with and without demographic controls, respectively) report on how the actual author (representing accessibility) and the reported author (representing signaling) affect our measures of perceptions of quality. The figures on the left hand side are means across the four experimental arms and the figures on the right hand side are generated from Equation (1). Panel a) reports the effects of the treatments on the unweighted average quality across the 10 attributes, Panel b) reports the weighted average across 9 attributes that respondents stated they found important, and, as a robustness check, Panel c) reports the unweighted average including only the 9 attributes. Red and blue lines are point estimates and 95% confidence intervals from regressions without and with controls, respectively.

Several patterns emerge. The left hand side of Panel (a) summarizes the main result of the paper. Blogs written by humans and truthfully disclosed as such (Human-Human) receive the highest ratings, followed by AI-AI, while AI-Human is rated lowest, already suggesting a preference for truthful disclosure. AI-generated blogs receive lower quality ratings, with one coefficient significant with controls and all coefficients negative and all statistically significant at the 10% level without controls (using the RI *p*-values), reflecting a 5–6% difference – a relatively small effect. Robustness checks using alternative measures confirm this pattern, with two additional measures showing statistical significance at the 5% level, and one of those showing a 12% difference. This suggests either a penalty for ambiguity or genuinely lower AI-generated quality. Notably, the reported author being AI has no significant impact on perceived quality.

Second, the negative perception of AI-generated blogs is more than offset when the author is correctly disclosed as AI, with statistical significance at the 10% level using the RI *p*-values for two of the three measures (and again an additional two indices significant at the 5% level) with controls.

Going forward, we refer to specifications that include controls and to the RI *p*-values in interpreting significance. All tables are presented in the Appendix. Each set of results is organized as follows: outcome indices with controls, outcome indices without controls, disaggregated components of the indices with controls, and disaggregated components without controls.

In exploratory analysis, we test potential mechanisms via heterogeneity analyses using the unweighted quality average across the 10 attributes as the outcome. We first note that given our relatively small sample size, statistical power could be limited. Ex-post power calculations suggest that we are reasonably powered in the quality outcomes, and thus, to the extent that we may be sufficiently powered to detect some differential effects, we attempt such an analysis.

First, to assess whether AI blogs are indeed lower in quality, we examine accessibility via the Flesch-Kincaid grade level [19]. AI-generated blogs are more "accessible," with a lower grade level (14.6 vs. 16.9), fewer words per sentence (20 vs. 27), and fewer paragraphs (12.6 vs. 13.8), though more sentences per paragraph (3.6 vs. 2.2). However, interacting the main coefficients with grade level shows no significant effect on perceived quality (see Fig 2 Panel a) and Appendix Tables S5 to S8 in S1 File). The only exception is a statistically significant effect on 'clear rationale', but since aggregated measures show no effects, we do not emphasize this result.

Appendix Tables S9 and S10 in S1 File show how experimental groups rated individual quality measures, our pre-specified secondary outcomes. AI-generated blogs scored lower on appropriate length and detail, visual presentation, tone, and ease of understanding when the true author was not disclosed. Differences between AI generated blogs that were disclosed as such are present for: title sparked curiosity (7% difference) and visual presentation (14% difference), significant at the 10% level in specifications with controls. These are still relatively small magnitudes. Notably, visual presentation was made identical across AI and human-written blogs, and length differences were minimal. Unlike [17], our highly educated sample (80% with a master's or higher and with technical expertise in the subject matter) may view accessibility differently, favoring technical language and detail.

Respondents' perceptions may also depend on prior AI use [42,58]. In our sample, 19% of respondents had not heard of AI, but among those aware of it, 76% had used an AI tool. Ambiguity may be higher for non-users, making it harder for them to assess AI-generated content and the quality of the underlying research. AI-generated blogs were rated

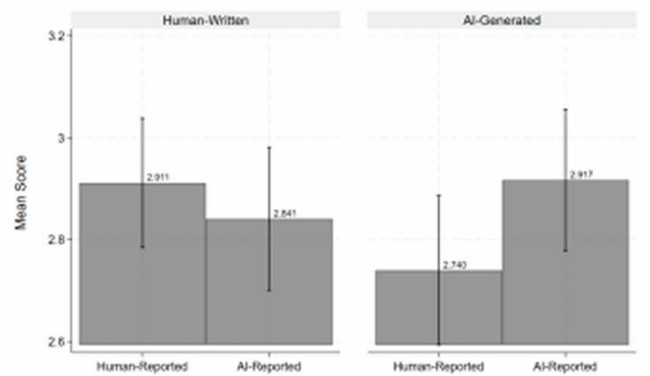 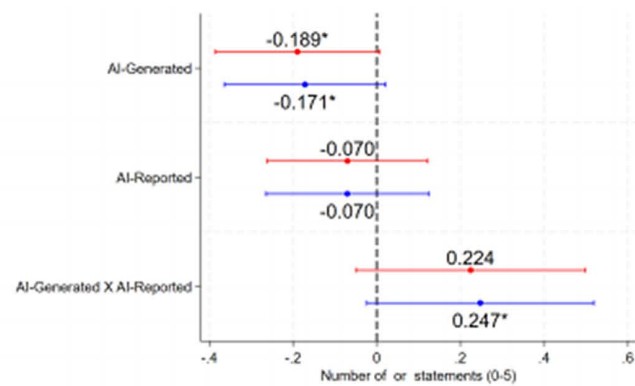

**(a)** Unweighted Average (of 10 attributes)

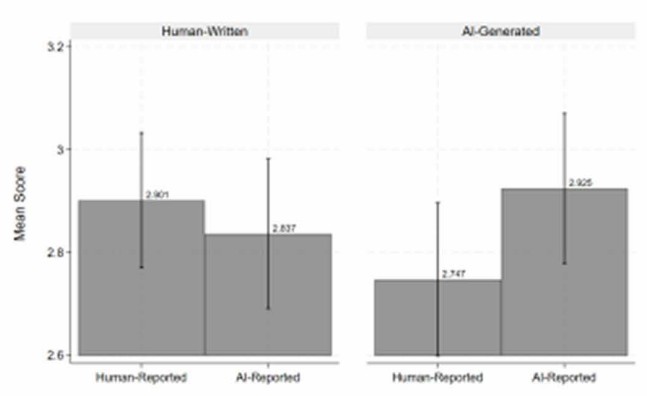 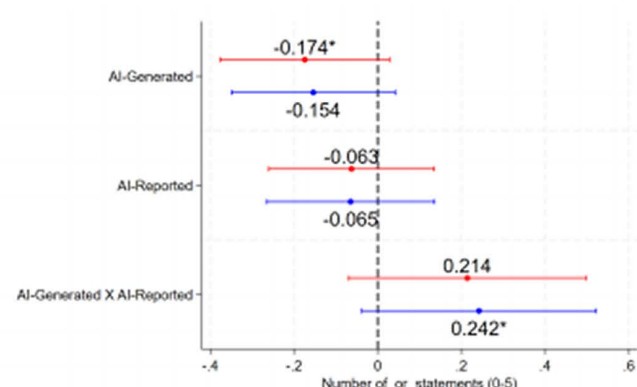

**(b)** Weighted Average (of 9 attributes)

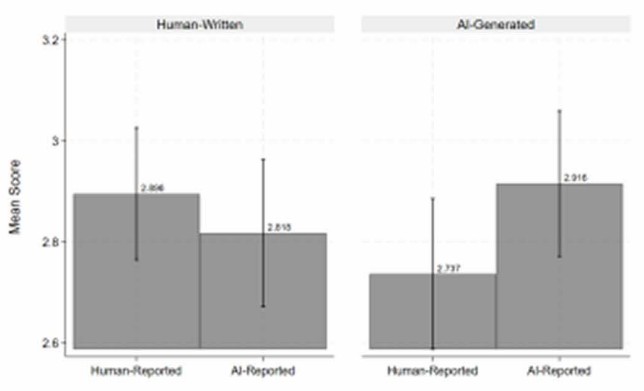 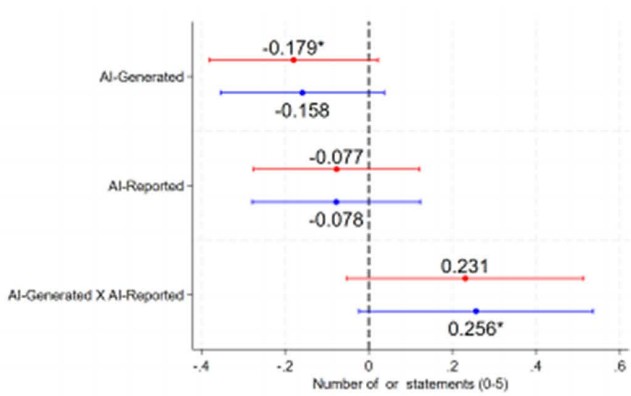

**(c)** Unweighted Average (of 9 attributes)

**Fig 1. Perceptions of the Quality of the Blogs Fig 1.** * *Notes*: * $p < 0.1$, ** $p < 0.05$, *** $p < 0.01$. This figure reports on the perceptions of blog quality measured using 10 attributes of high quality blogs [45,52–54]. Red and blue lines are coefficients and 95% confidence intervals from regressions without and with controls, respectively. Perceptions are measured using a 5-point Likert scale ranging from 'strongly disagree' (0) to 'strongly agree' (4) that the blog is high-quality with respect to that attribute. The 10 attributes are: blog had an appropriate amount of detail, a clear rationale, a catchy title, good visual presentation, an appropriate tone, was of adequate length, was easy to understand, and provided clear, relevant, and sufficiently detailed

recommendations. The outcome in Panel a) is the simple average across 10 blog attributes, and the outcome in Panel b) is the weighted average across 9 attributes (weighted by respondents' reported relative importance of attributes – tone was not rated). The outcome in Panel c) is the unweighted average for the same 9 attributes as in Panel b) for robustness. Means are reported on the left hand side and coefficients from a regression of the measure of quality on indicator variables for AI-generated blogs, AI-reported blogs, and their interaction are reported on the right hand side. We control for wave and strata (country, seniority, and gender) fixed effects. Standard errors are clustered at the individual level (the unit of randomization).

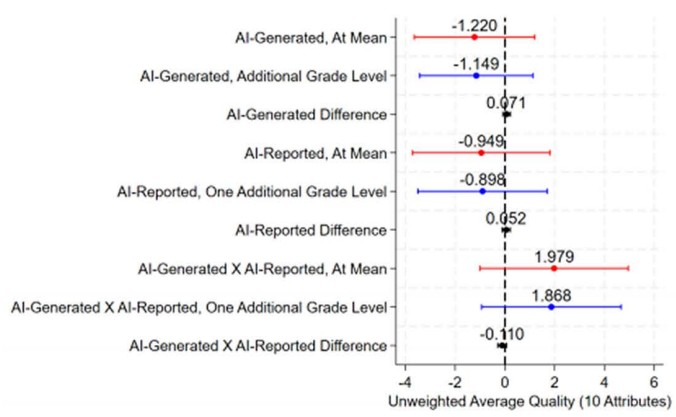

(a) Flesch-Kinkaid Reading Level

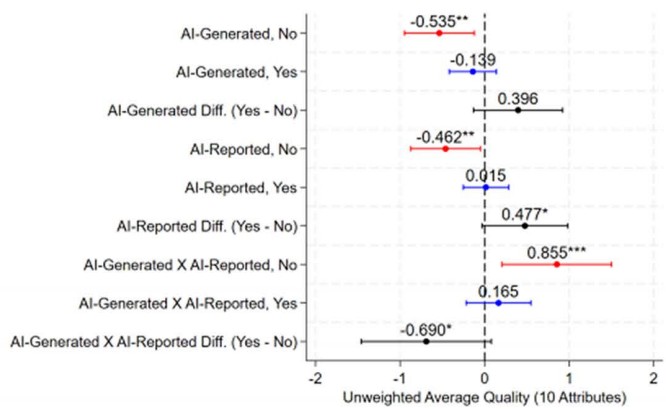

(b) Yes = Used AI

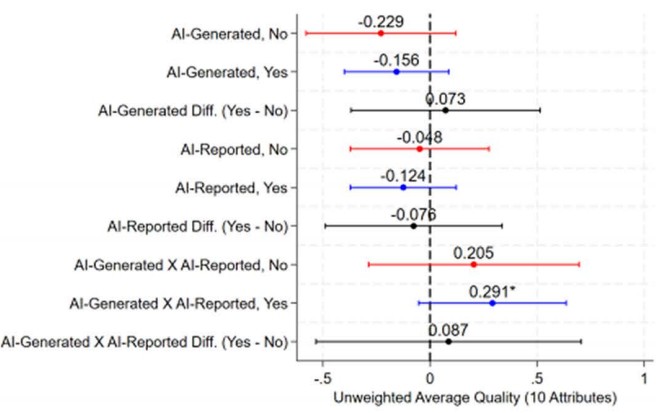

(c) Yes = Trust AI

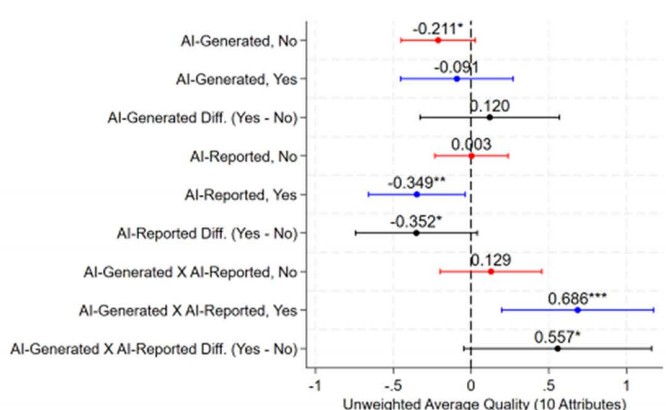

(d) Yes = Female

**Fig 2. Heterogeneity in Perceptions of the Quality of the Blogs Fig 2.** * *Notes*: * p<0.1, ** p<0.05, *** p<0.01. This figure reports on the perceptions of blog quality measured using 10 attributes of high quality blogs [45,52–54] across four dimensions of heterogeneity. Panel a) reports on the grade level of the blog, Panel b) on whether the respondent has previously used AI, Panel c) on whether the respondent trusts AI to write high-quality blogs, and Panel d) by gender (female=1). Red (attribute=0), blue (attribute=1), and black (difference) lines are point estimates and 95% confidence intervals. Perceptions are measured using a 5-point Likert scale ranging from 'strongly disagree' (0) to 'strongly agree (4) that the blog is high-quality with respect to that attribute. The 10 attributes are: blog had an appropriate amount of detail, a clear rationale, a catchy title, good visual presentation, an appropriate tone, was of adequate length, was easy to understand, and provided clear, relevant, and sufficiently detailed recommendations. We control for respondent characteristics, wave, and strata (country, seniority, and gender) fixed effects. Standard errors are clustered at the individual level (the unit of randomization).

approximately 0.5 points lower by non-users ($p<0.05$) and 0.14 points lower by users, though the difference was not statistically significant. However, non-users penalized AI-reported authorship more (0.5 points lower, $p<0.05$). This result could point to a behavioural bias against AI as this unfamiliarity increases ambiguity. When AI-generated blogs were disclosed as such, non-users rated them 0.9 points higher, with a 0.7-point gap between users and non-users (significant at the 10% level). These magnitudes are larger at about 23%.

We also examine the role of trust in AI, as explored by [42], which may influence the perceived value of a communication. In our sample, 60% of respondents reported trusting AI to write high-quality blogs. When interacting this indicator (trust AI = 1) with the treatment arms, we find that respondents who trust AI rate blogs more highly overall and when the author is disclosed as AI, but trust does not significantly differentially affect perceived quality (see Fig 2 Panel c) and Appendix Tables S15 to S18 in S1 File). Disaggregated results show a similar pattern for appropriate level of detail and visual presentation in the specifications with controls. Trust in AI may not help readers assess the quality of underlying research, so disclosure may not differentially mitigate ambiguity effects between those who trust AI and those who do not.

Finally, we examine gender-based heterogeneity, a pre-specified analysis given some evidence that gender influences perceptions about AI [59], with privacy and potential risks explaining a quarter of the gender gap in AI use. As shown in Fig 2 Panel d), women penalize AI-reported blogs more than men but also rate truthfully disclosed AI-generated blogs more highly ($p<0.05$). There is a statistically significant ($p<0.05$) difference between men and women when AI authorship is disclosed (see also Appendix Tables S19 to S22 in S1 File that show that women tend to rate blogs as lower quality in general and that the the title, visual presentation, and ease of understanding show small significant results). Note that we find that women are ten percentage points less likely to report trusting AI compared to men ($p=0.067$), suggesting that trust may play a role.

These results all support the conjecture that the respondents in this sample place more value on signaling, preferring an unambiguous signal to supposed lower effort since they may already trust and use AI.

## Engagement

We next discuss engagement with the blogs. The left hand side panels of Figs 3 and 4 show that AI-generated blogs have lower reported intended levels of engagement and beliefs about others' engagement, regardless of who the reported author is. The AI-generated blogs that are truthfully disclosed as such are rated slightly higher compared to those reported as having been written by a human.

These figures and Appendix Tables S23 to S26 in S1 File show no statistically significant effects of any treatment on respondents' intended engagement, aggregated or by dimension, or on time spent reading the blog. Similarly, there are no significant effects on beliefs about others' engagement (see Tables S27 to S30 in S1 File).

There may be several dimensions across which levels of engagement may vary. The respondent may or may not be the one who actually makes the policy decisions (they may not be senior enough or even if they are, they still may lack policy influence), those who rated the blog low in quality may engage less, and there may be differences by gender. We check each of these dimensions of heterogeneity with respect to respondents' own and others' engagement and find no strong evidence of any differences by seniority, policy influence, or the respondent's quality rating (see Supporting Information S1 and S2 Figs and Appendix Tables S31 to S54 in S1 File). As expected, respondents who rated the blog as high quality reported both higher levels of intended engagement, and perceived engagement by others overall especially when the blog is reported as AI-generated.

However, again, gender exhibits some differences with women reporting lower intended engagement and the differences between men and women when AI use is disclosed as such being statistically significant at the 5% level and quite large at between 34–50% (there is no impact on time spent reading the blog, however). These effects are most pronounced for sharing the blog with others, re-reading the blog, and looking up studies cited (see Appendix Tables S55 to S58 in S1 File). In terms of others' engagement, women again rate the likelihood of others' engagement lower in general

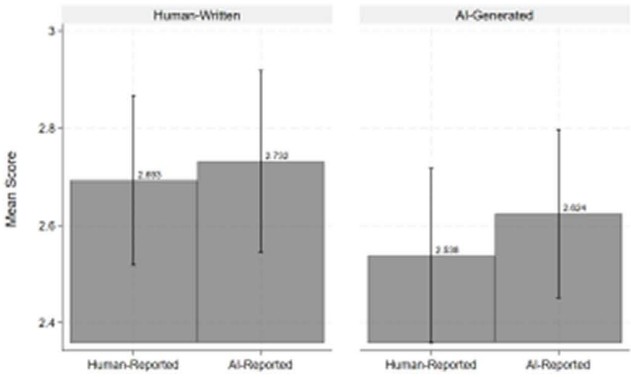
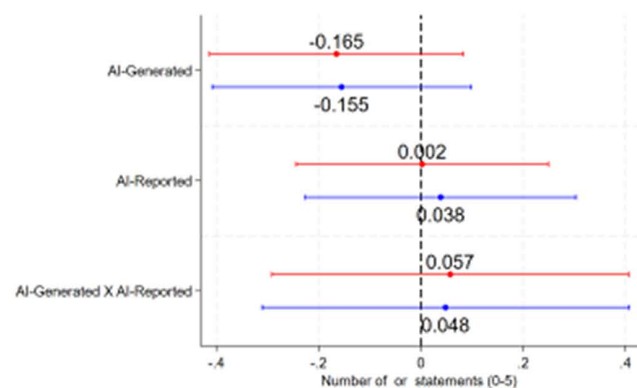

(a) Unweighted Average (0-4)

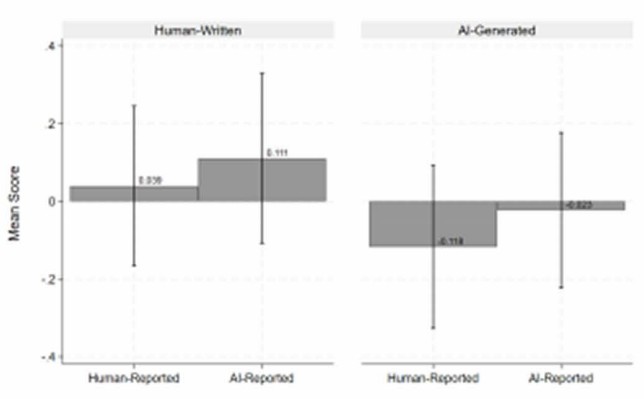
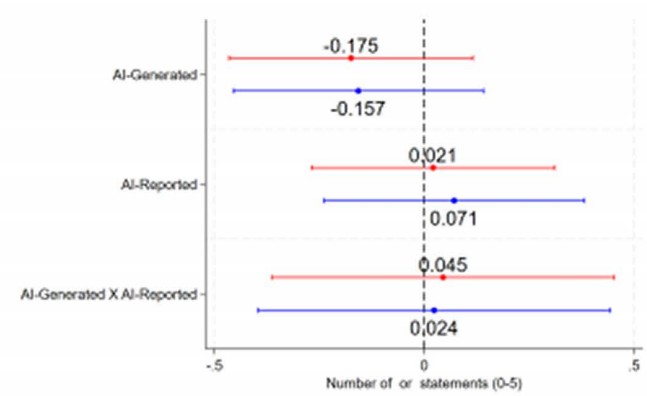

(b) Standardized First Principal Component

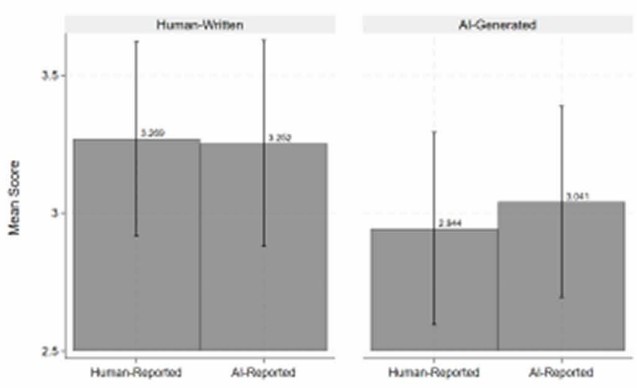
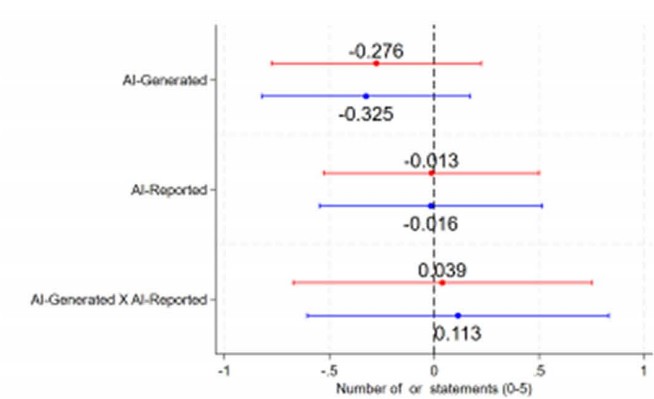

(c) Number of 'likely' and 'very likely' statements (0-5)

**Fig 3. Intended Engagement with the Blog Fig 3.** * *Notes*: * p<0.1, ** p<0.05, *** p<0.01. This figure reports on the likelihood of the respondents' engagement with the blog. Red and blue lines are coefficients from regressions without and with controls, respectively, and include the point estimate and 95% confidence interval. Engagement is measured across five actions: whether they are likely to re-read the blog, share the blog with others, look up studies cited in the blog, look up relate studies, or contact the authors. The responses follow a Likert scale ranging from 0 to 4, with 0 indicating 'very unlikely' and 4 indicating very likely'. The outcome in Panel a) is the average rating, Panel b) is the standardized score of the first principal component, and Panel c) is the number of actions where the respondent indicated they are 'likely' or 'very likely' to take that action. We control for wave and strata (country, seniority, and gender) fixed effects. Standard errors are clustered at the individual level (the unit of randomization).

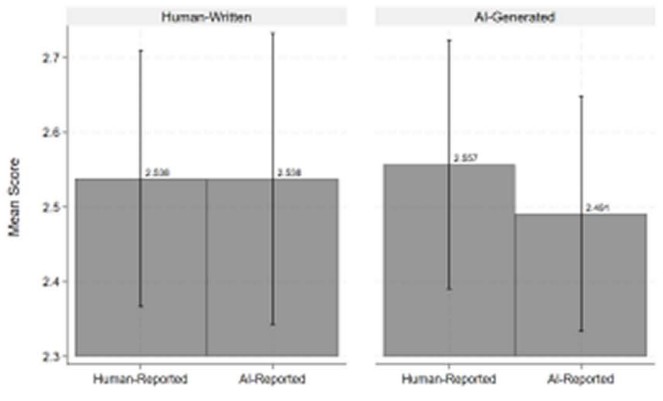
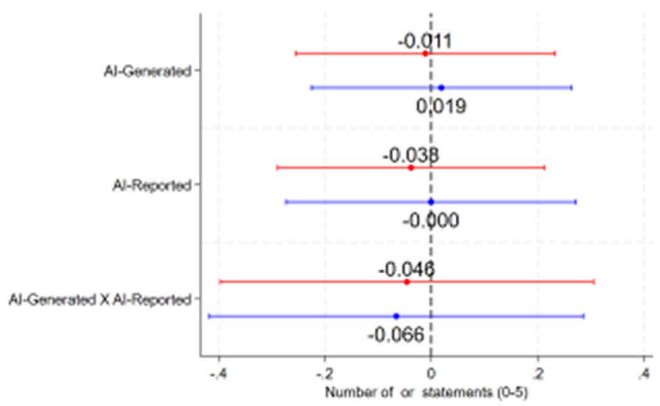

(a) Unweighted Average (0-4)

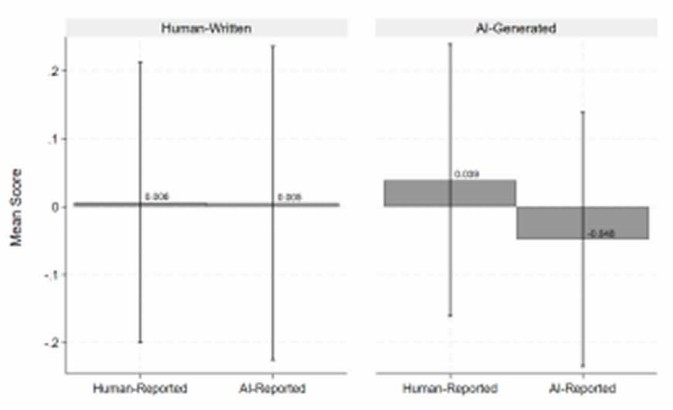
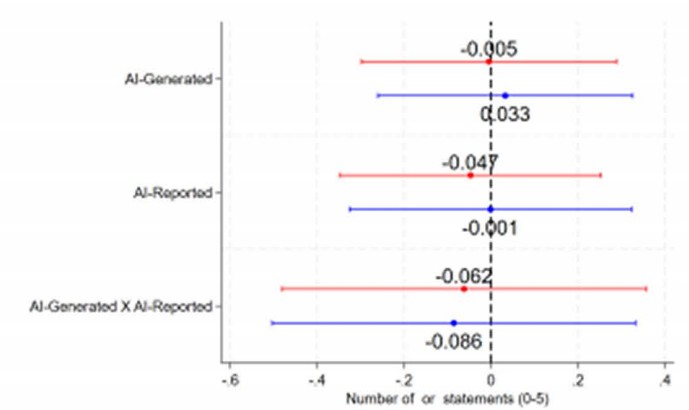

(b) Standardized First Principal Component

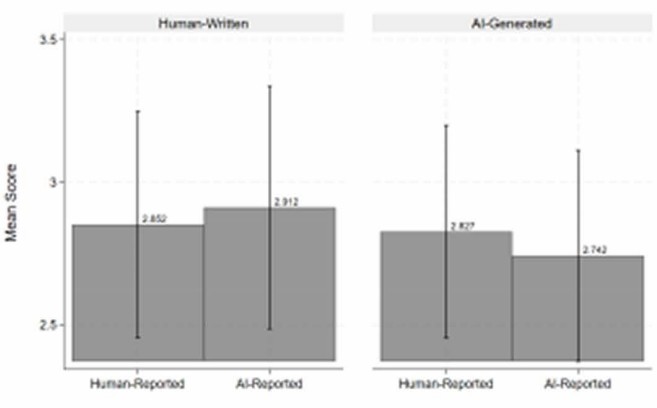
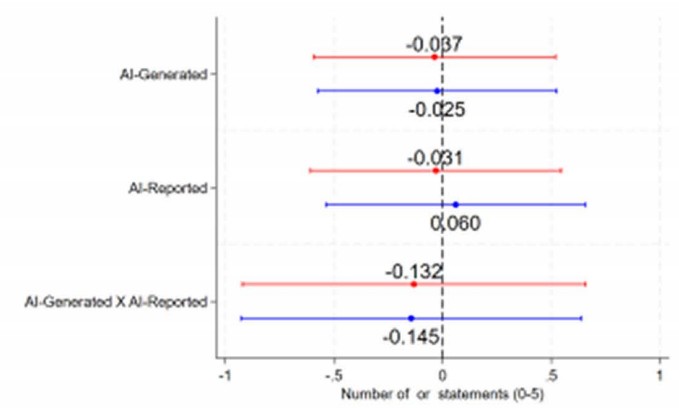

(c) Number of 'likely' and 'very likely' statements (0-5)

**Fig 4. Beliefs About Others' Intended Engagement with the Blog Fig 4.** * *Notes*: * $p<0.1$, ** $p<0.05$, *** $p<0.01$. This figure reports on the respondents' beliefs about others' likelihood of engagement with the policy brief. Red and blue lines are coefficients from regressions without and with controls, respectively, and include the point estimate and 95% confidence interval. Engagement is measured across five actions: whether respondents believe that others are likely to re-read the blog, share the blog with others, look up studies cited in the blog, look up relate studies, or contact the authors. The responses follow a Likert scale ranging from 0 to 4, with 0 indicating 'very unlikely' and 4 indicating very likely'. The outcome in Panel a) is the average rating, Panel b) is the standardized score of the first principal component, and Panel c) is the number of actions where the respondent indicated that

others are 'likely' or 'very likely' to take that action. We control for wave and strata (country, seniority, and gender) fixed effects. Standard errors are clustered at the individual level (the unit of randomization).

and when the reported author is AI, but rate it much higher when the use of AI is correctly disclosed; these magnitudes are equally high and consistent across the different dimensions of engagement (see Appendix Tables S59 to S62 in S1 File). Note that 77% of men and 62% of women report previous use of AI, so there may be some overlap in interpretation of the results regarding gender and the use of AI, but we do not believe this would be large enough to drive the results on gender.

## Discussion

The main implication of this paper is that academics and other technical writers can credibly use AI to convey their research in a more accessible way, under certain conditions. In this study, since neither the actual nor reported author substantially affects respondents' engagement and *if* a disclosure statement is provided, AI-generated communications are not viewed as lower quality. AI may be a viable alternative for researchers when writing blogs for technical audiences, since it reduces their cost of effort without compromising perceptions of quality or any potential engagement, on average. However, there may be cases where the audience is less receptive to AI or where a premium is placed on human-generated content.

We innovate by using a randomized experiment comparing the actual and reported authorship of research blogs (AI vs. Human) whereby respondents from eleven countries rated a blog's quality. AI-generated content was rated lower in quality than human-written content. However, when an AI disclosure statement was included, respondents did not penalize AI-generated material, highlighting the importance of truthful signaling of authorship in research communication. Engagement with, and others' predicted engagement with the blogs were not affected by the true or reported author.

That the negative perception of quality of AI-generated blogs is offset by truthful disclosure suggests that the respondents in this sample slightly penalize ambiguity and that the advantage of eliminating ambiguity outweighs the penalty of exposing the potentially lower effort involved in using AI. Consequently, researchers can benefit, on average, from the reduced cost of effort of using AI in research blogs without incurring the penalty in perceptions of quality of signaling low effort *provided* they include a disclosure statement.

That this sample's preferred blog length exceeded the actual length suggests that preferences regarding accessibility depend on the audience. Results by the grade level of the blog suggest that a lower reading level may also signal weaker research quality, as academic language often conveys rigor. Researchers should thus tailor their AI prompts, and/or the decision to use AI, to their audiences accordingly.

Since users who had not previously used AI rated the blogs more positively compared to those who had previously used AI when true authorship was disclosed suggests that eliminating ambiguity may benefit non-users more, with disaggregated results indicating that the effects are primarily driven by appropriate tone and clear, relevant recommendations in the specifications with controls.

Our findings on gender align with those of [42], who also found that prior AI use does not predict perceptions of quality by whether authorship information is provided. Unlike [42], however, we find a statistically significant (1% level) but small (0.17) correlation between AI use and trust in AI to write high-quality blogs.

Beyond its role in reducing informational ambiguity, disclosure of AI use also relates to ongoing debates in research ethics. While our experiment was not designed to evaluate ethical norms, the findings suggest an important alignment: transparent disclosure does not appear to carry a perceived quality or engagement penalty in this context and may mitigate negative perceptions associated with ambiguity. At the same time, our results do not shed light on broader ethical concerns surrounding AI use in research, including authorship credit, responsibility for errors, or the preservation of scholarly norms.

The study has two key strengths. The first is the unique design we were able to implement – namely, the presence of existing blogs written by humans and their associated academic papers from which we could prompt the writing of an AI blog. Additionally, we have access to a sample for whom this type of information is arguably extremely useful, rather than the general public. However, the study also has some limitations. First, while the sample is relevant and lends internal validity, it is a convenience sample and does not represent all types of users who would find the type of evidence produced relevant; our sample is highly educated and connected to research and policy and does not include journalists, for example. In fact, the demographics of our sample, particularly high levels of education, is likely the most important driver of our result. Consequently, these results are most applicable for researchers who would like to communicate their research to stakeholders who are technically advanced compared to the general public. An additional limitation is our measure of "readability." We note that the Flesch-Kincaid model was developed for English speakers in the US and may not be as good of a measure in contexts where English is not the first language. Finally, the outcomes are also self-reported, potentially rendering them over- or under-reported.

Further investigation of mechanisms, particularly further study of the role of gender, would add to our understanding of how to leverage AI in research. Additionally, other types of writing could be studied, for example, for more general audiences, and samples from high-income countries and different demographic groups would also provide important evidence, particularly on how the engineering of prompts could potentially be improved. Future work could also incorporate actual behaviour rather than self-reported intended behaviour. For example, providing the opportunity for respondents to click through to the underlying paper or to the authors' websites and tracking this behaviour. While in this sample we find that AI is viewed positively, caution is advised when using AI, as it is still prone to errors, trust in AI is still relatively low in some contexts, and the field is rapidly evolving.

## Supporting information

**S1 Fig. Heterogeneity in Intended Engagement with the Blog.** *Notes*: * $p < 0.1$, ** $p < 0.05$, *** $p < 0.01$. This figure reports on the respondents' reported likelihood of engagement across four dimensions of heterogeneity. Engagement is measured as the mean of five actions: whether they are likely to re-read the blog, share the blog with others, look up studies cited in the blog, look up relate studies, or contact the authors. The responses follow a Likert scale with 0 indicating that they are 'very unlikely' and 4 indicating they are 'very likely' to take that action. Panel a) is whether the respondent holds a senior position, Panel b) is whether the respondent holds a high degree of influence in policy-making, Panel c) is whether the respondent gave the blog a 'high' or 'very high' quality rating, and Panel d) is whether the respondent is female. We control for respondent characteristics, wave, and strata (country, seniority, and gender) fixed effects. Standard errors are clustered at the individual level (the unit of randomization).
(TIFF)

**S2 Fig. Heterogeneity in Beliefs about Others' Intended Engagement with the Blog.** *Notes*: * $p < 0.1$, ** $p < 0.05$, *** $p < 0.01$. This figure reports on the respondents' reported beliefs about others' likelihood of engagement across four dimensions of heterogeneity. Engagement is measured as the mean of five actions: whether others are likely to re-read the blog, share the blog with others, look up studies cited in the blog, look up relate studies, or contact the authors. The responses follow a Likert scale with 0 indicating that others are 'very unlikely' and 4 indicating they are 'very likely' to take that action. Panel a) is whether the respondent holds a senior position, Panel b) is whether the respondent holds a high degree of influence in policy-making, Panel c) is whether the respondent gave the blog a 'high' or 'very high' quality rating, and Panel d) is whether the respondent is female. We control for respondent characteristics, wave, and strata (country, seniority, and gender) fixed effects. Standard errors are clustered at the individual level (the unit of randomization).
(PDF)

**S1 File. Appendix.**
(PDF)

## Acknowlegment

This trial was registered with the American Economics Association, registry number AEARCTR-0012495.

## Author contributions

**Conceptualization:** Michael Keenan.

**Data curation:** Jawoo Koo, Christine Mwangi, MinAh Kim.

**Formal analysis:** Michael Keenan, Naureen Karachiwalla, Christine Mwangi.

**Investigation:** Michael Keenan, Naureen Karachiwalla.

**Methodology:** Michael Keenan, Naureen Karachiwalla, Jawoo Koo, MinAh Kim.

**Project administration:** Michael Keenan, Clemens Breisinger.

**Resources:** Clemens Breisinger.

**Software:** Michael Keenan, Jawoo Koo, Christine Mwangi, MinAh Kim.

**Supervision:** Michael Keenan, Naureen Karachiwalla, Clemens Breisinger.

**Visualization:** Michael Keenan.

**Writing – original draft:** Michael Keenan, Naureen Karachiwalla.

**Writing – review & editing:** Michael Keenan, Naureen Karachiwalla, Christine Mwangi.

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
