## [Decision Letter · Decision Letter 0]

22 Sep 2025

Dear Dr. Keenan,

Thank you for submitting your manuscript to PLOS ONE. After careful consideration, we feel that it has merit but does not fully meet PLOS ONE’s publication criteria as it currently stands. Therefore, we invite you to submit a revised version of the manuscript that addresses the points raised during the review process.

We look forward to receiving your revised manuscript.

Kind regards,

Jafar Kolahi

Academic Editor

PLOS ONE

Journal Requirements:

This work was supported by the CGIAR Initiatives on National Policies and Strategies (NPS) and Digital Innovation (DI) and the Bill & Melinda Gates Foundation through the Generative AI for Agriculture project (INV-047346).

6. Please amend your list of authors on the manuscript to ensure that each author is linked to an affiliation. Authors’ affiliations should reflect the institution where the work was done (if authors moved subsequently, you can also list the new affiliation stating “current affiliation:….” as necessary).

8. Please remove all personal information, ensure that the data shared are in accordance with participant consent, and re-upload a fully anonymized data set.

Additional guidance on preparing raw data for publication can be found in our Data Policy (https://journals.plos.org/plosone/s/data-availability#loc-human-research-participant-data-and-other-sensitive-data) and in the following article: http://www.bmj.com/content/340/bmj.c181.long .

Additional Editor Comments:

Title is too broad. It does not specify that the medium studied is blogs nor that the design is cross-country survey experiment.

Extent the abstract section and summarize aim, methods, results, and discussion (without subheadings).

In several places, the manuscript cites references by number only (e.g., “the seminal paper by [15]”). Author names should be included where appropriate, e.g., “The seminal paper by Akerlof GA [15].”

Methodological details were spread between the Introduction and the Experiment Design section. Please consolidate into a standalone Methods section, following standard IMRAD structure. This section should include study design and population, outcomes measured and statistical analysis plan.

Specify the method, algorithm, or tool employed for randomization.

Which software used for data analysis? Add with details at the Methods section.

AI-generated blogs were rated lower in quality despite being more accessible I suggest examining potential bias against AI alongside textual features (e.g., tone, jargon, stylistic markers) where AI outputs may differ from human blogs. Such analysis could clarify why readers perceive AI texts as lower quality.

Consider including one figure that directly summarizes the main result e.g., a concise visualization comparing perceptions of quality across the four experimental arms.

I recommend separating out a full Discussion section. This will allow a fuller exploration of:

• Interpretation of findings.

• Limitations (e.g., highly educated sample, external validity).

• Implications for research communication and directions for future research.

Reviewers' comments:

Reviewer's Responses to Questions

**Comments to the Author**

1. Is the manuscript technically sound, and do the data support the conclusions?

Reviewer #1: Partly

Reviewer #2: Yes

Reviewer #3: Yes

2. Has the statistical analysis been performed appropriately and rigorously?

Reviewer #1: Yes

Reviewer #2: Yes

Reviewer #3: Yes

3. Have the authors made all data underlying the findings in their manuscript fully available?

Reviewer #1: Yes

Reviewer #2: Yes

Reviewer #3: Yes

4. Is the manuscript presented in an intelligible fashion and written in standard English?

Reviewer #1: Yes

Reviewer #2: Yes

Reviewer #3: Yes

Reviewer #1: General Comments

This manuscript presents a well-designed 2x2 factorial experiment investigating how AI-generated versus human-written research blogs, and the disclosure of their authorship, affect perceptions of quality and engagement among research consumers. The study is timely, addresses a highly relevant topic in academic communication, and is grounded in a solid theoretical framework of information asymmetry. The methodology is clearly described, including the pre-registration and IRB approval, and the statistical analysis is appropriate. The paper is exceptionally well-written and logically structured.

However, the study has two major weaknesses that require substantial revision, along with a few minor points that would further strengthen the manuscript.

Major Weaknesses:

Sample Generalizability: The manuscript's primary limitation is its reliance on a convenience sample drawn from IFPRI mailing lists and social media, which consists of highly educated specialists (over 80% with a master's degree or higher). While the authors acknowledge this , the paper's framing and language frequently generalize the findings to broader populations (e.g., "readers," "the informed general public"). The study's conclusions are only directly applicable to a very specific, elite group of research consumers. The manuscript must be systematically revised to consistently frame the context, interpretation, and implications of the findings around this specific demographic. This should be explicitly stated in the Abstract, Methods, Discussion, and Conclusion to avoid overstating the external validity of the results.

Interpretation of Accessibility: The study uses the Flesch-Kincaid grade level as an "objective measure of accessibility" and finds that AI-generated blogs are more "accessible" (i.e., have a lower grade level). However, for the highly specialized audience surveyed, a lower reading level may not be perceived as a positive attribute. It could instead signal a lack of technical nuance, oversimplification, or lower scientific rigor. The assumption that greater simplicity equates to higher quality is a critical conceptual issue for this specific sample. The Discussion section should more deeply explore this alternative interpretation as a primary mechanism for the findings, rather than just a penalty for ambiguity.

Minor Weaknesses:

Justification for Statistical Model: The analysis relies on OLS for ordinal outcomes (Likert scales). While this is common practice, a brief justification for this choice over models designed for ordinal data (e.g., ordered logit) would strengthen the statistical methodology section. A robustness check using such a model could also be beneficial.

Cross-Cultural Validity of Readability Metric: The Flesch-Kincaid score was developed for American English. Its application across a diverse sample from 11 countries, where English is often not the first language, is a limitation. While acknowledged, this point could be briefly noted in the methodology section as a potential source of measurement error.

Power for Subgroup Analyses: The heterogeneity analyses, particularly concerning gender, are intriguing . However, with a total sample of 366, the statistical power for these subgroup comparisons may be limited. A brief comment on the exploratory nature of these findings or a post-hoc power analysis would be appropriate.

Specific Comments

Page 1, Abstract: The phrase "readers rate the quality" should be specified to better reflect the sample, for instance, "research stakeholders rate the quality...". This clarifies the population from the outset.

Page 11, Section 3.1, Paragraph 3: The statement that the online survey was distributed to reach a "variety of stakeholders" should be more precise. The description that follows indicates a rather homogenous group of highly educated professionals in specific fields. Please rephrase to accurately reflect the sample's characteristics.

Page 12, Section 4.2, Paragraph 1: The statement "truthful disclosure of AI use offsets the negative effect more for women and those who have used AI" is a key finding from the heterogeneity analysis. Given the potential power issues mentioned in the general comments, this claim should be presented with a degree of caution.

Page 19, Section 3.3, Paragraph 4: Regarding the Flesch-Kincaid score, please add a sentence explicitly acknowledging the limitations of applying this US-centric, English-language metric to a diverse, multi-national, non-native English-speaking sample.

Page 22, Section 4.2, Paragraph 4: The discussion posits that the highly educated sample "may view accessibility differently, favoring technical language and detail". This is a crucial point and likely a primary driver of the results. This argument should be elevated from an exploratory point to a central theme in the Discussion section. The negative perception of the lower-grade-level AI text may not be a penalty for ambiguity but a correct assessment of its lower value to this specific audience.

Page 25, Section 4.2, Paragraph 4: The gender analysis reveals that women "rate truthfully disclosed AI-generated blogs more highly (p<0.05)". This is a fascinating result. The manuscript would benefit from a more developed theoretical interpretation of this finding. Why might women value the truthful signal of AI authorship more than men in this context?

Recommendations for Additional Citations

Page 10, Introduction, Paragraph 4: To strengthen the discussion on the scientific community grappling with the responsible use of AI, consider including literature that reflects on the broader ethical and practical dilemmas AI poses to core academic processes like peer review.

[Ben Saad H, Dergaa I, Ghouili H, Ceylan Hİ, Chamari K, Dhahbi W: The assisted Technology dilemma: a reflection on AI chatbots use and risks while reshaping the peer review process in scientific research. AI & SOCIETY 2025:1-8.]

Page 13, Introduction, Paragraph 5: When discussing the main use-cases of AI for economic research, such as idea generation and summarizing text, it would be beneficial to cite a review that illustrates the breadth of AI applications in another specific scientific domain. This would exemplify its role beyond communication and into areas like motion analysis and injury prevention.

[Souaifi M, Dhahbi W, Jebabli N, Ceylan Hİ, Boujabli M, Muntean RI, Dergaa I. Artificial Intelligence in Sports Biomechanics: A Scoping Review on Wearable Technology, Motion Analysis, and Injury Prevention. Bioengineering. 2025; 12(8):887.]

Page 14, Conceptual Framework, Paragraph 4: In the discussion of readers' "cognitive effort and time investment," you could briefly contextualize the importance of cognitive performance by citing an example from another field that demonstrates how even simple physiological interventions can significantly impact cognitive outcomes, reinforcing the value of reducing cognitive load for the reader.

[Bouzouraa E, Dhahbi W, Ferchichi A, Geantă VA, Kunszabo MI, Chtourou H, Souissi N. Single-Night Sleep Extension Enhances Morning Physical and Cognitive Performance Across Time of Day in Physically Active University Students: A Randomized Crossover Study. Life. 2025; 15(8), 1178.]

Page 13, Introduction, Paragraph 4: When discussing the need for responsible use and disclosure of AI, you could support this point by drawing a parallel to the broader scientific need for rigorous validation of any new technology or device before its widespread adoption, citing an example of such a validation study from the sports science literature.

[Ardigò LP, Palermi S, Padulo J, Dhahbi W, Russo L, Linetti S, Cular D, Tomljanovic M: External responsiveness of the SuperOpTM device to assess recovery after exercise: A pilot study. Frontiers in sports and active living 2020, 2:67.]

Page 23, Results, Paragraph 1: In the exploratory analysis where you discuss biases in perceptions about AI, you could draw a brief analogy to other fields where subjective perception is a key driver of behavior, such as how taste perception influences dietary intake, to frame the broader relevance of studying perceptual biases.

[Karmous I, Ben Othman R, Dergaa I, Ceylan Hİ, Bey C, Dhahbi W, Sayed KA, Jamoussi H, Ioan MR, Khan NA. Sweet and Fat Taste Perception: Impact on Dietary Intake in Diabetic Pregnant Women—A Cross-Sectional Observational Study. Nutrients. 2025;17(15), 2515.]

Reviewer #2: The manuscript studies how readers judge short research blogs when authorship is either human or AI and when the label shown to readers either tells the truth or misleads. Across an eleven country sample and a preregistered two by two experiment, the authors report that AI generated posts receive slightly lower quality ratings than human written posts, yet this difference disappears when readers are truthfully told that a post was produced by AI. Readability, measured with standard indices, is higher for the AI versions, while stated intentions to engage with the underlying research do not differ across conditions. The question is timely, the design is clean, and the results are easy to apply for scholars who are deciding whether or how to disclose AI assistance.

The study’s contribution rests on three solid pillars. First, the authors pose a concrete, policy relevant dilemma that institutions and journals face now, and they frame it with an information and signaling logic that distinguishes ambiguity from honest disclosure. Second, the cross randomization of true authorship and displayed authorship neatly isolates that mechanism while preregistration, IRB approval, and clear outcomes bolster credibility. Third, the authors report heterogeneity by gender and by prior AI use, which invites practical segmentation for science communication. Taken together, these features make the paper readable, transparent, and directly useful for editorial and outreach policy.

The most important limitation concerns who the readers are. The sample is highly educated and connected to a policy research network, which is a strength for internal validity but narrows external validity. Such audiences may be unusually tolerant of technical prose, unusually familiar with AI tools, and unusually attentive to disclosure language. I encourage the authors to foreground this boundary condition in the abstract and opening section and to consider a replication on a general audience panel, in undergraduate populations who often consume research explainers, and in non English settings. Even a brief appendix table that compares demographics and AI use to national benchmarks would help readers judge generalizability.

A second limitation lies in the outcomes. The paper relies on Likert ratings and self reported intentions to read, share, or follow up, which are informative but do not capture behavior. If server side tracking is not feasible, the authors could add a short follow on task with incentive compatible choices, such as a chance to click through to the underlying article, to bookmark, or to trade earnings for more information. Even a simple measure of time on page or a binary click through outcome would substantively strengthen the claim that disclosure does not reduce engagement.

The presentation of stimuli and the theorized pathway could be elaborated. Readers need more detail on how the AI and human texts were generated and standardized: model and version, prompts, temperature, length constraints, topic balance, and any editing rules. Readability checks are helpful but do not exhaust style markers that shape perceived credibility. Simple text analytics on sentence length variance or lexical diversity would reassure that the two sources produced comparable prose. On the theory side, the signaling account could be tied more tightly to the observed heterogeneity. A compact path diagram that distinguishes disclosure effects that operate through expectations from those that operate through perceived clarity would clarify what the experiment identifies. Finally, with N spread across four arms and many countries, please report minimum detectable effects, country variance, and results with country fixed effects or clustering to calibrate precision.

Transparency and compliance details are mostly in order, but one item needs attention for this journal. Data and code should be available at publication. The paper currently states that data will be provided upon acceptance. Please deposit de identified data, code, survey instrument, preregistration link, and all blog stimuli in a trusted repository and update the data availability statement. With those improvements, my overall view is positive. I recommend revise and resubmit at the level of minor to moderate changes. The practical message will be valuable to editors, communicators, and researchers alike: one can use AI to produce accessible research summaries and, provided disclosure is honest and clear, expect no meaningful penalty in perceived quality or stated engagement.

To deepen your framing of audience priors, trust, and disclosure effects, please engage with recent higher education research on generative AI adoption from Palestinian and regional contexts. These studies clarify why perceptions differ by prior AI use and offer constructs you can mobilize (performance expectancy, effort expectancy, social influence, facilitating conditions, habit, anxiety). They will also broaden the geographic scope of your literature review and sharpen your policy implications.

1. Ayyoub, A., Khlaif, Z. N., Hamamra, B., Bensalem, E., Mitwally, M., Sanmugam, M., ... & Khaldi, M. (2025). Drivers of acceptance of generative AI through the lens of the extended Unified Theory of Acceptance and Use of Technology. Human Behavior and Emerging Technologies, 2025(1), 6265087.

2. Hamamra, B., Khlaif, Z. N., & Mayaleh, A. (2025). Between promise and precarity: Palestinian university educators’ experiences with AI integration in higher education. Interactive Learning Environments, 1–16.

3. Hamamra, B., Khlaif, Z. N., Mayaleh, A., & Baker, A. A. (2025). A phenomenological examination of educators’ experiences with AI integration in Palestinian higher education. Cogent Education, 12(1), 2526435.

4. Khlaif, Z. N., Hamamra, B., Bensalem, E., Mitwally, M. A., & Sanmugam, M. (2025). Factors influencing educators’ technostress while using generative AI: A qualitative study. Technology, Knowledge and Learning, 1–26.

These sources provide theory backed constructs for your mechanism section, empirics from a Global South context to strengthen external validity claims, and concrete variables you could measure in a follow up (acceptance, anxiety, technostress) to connect disclosure to downstream behavior. I recommend citing them in the related work when introducing audience priors and in the discussion when articulating policy and practice implications for research communication and higher education.

Reviewer #3: I would recommend acknowledging the limitations of the study and a stronger argumentation concerning further studies. Otherwise I consider the manuscript to be well written, presented in a logical and convincing manner, in good style and transparent with respect to the adopted methodology. I particularly highlight the fact that the study covers a solid number of countries and was conducted in four waves, ensuring the representativeness of the data.

**Do you want your identity to be public for this peer review?** For information about this choice, including consent withdrawal, please see our Privacy Policy

Reviewer #1: **Yes:** Wissem Dhahbi

Reviewer #2: No

Reviewer #3: **Yes:** Cernicova-Buca Mariana

---

## [Author Response · Author response to Decision Letter 1]

7 Nov 2025

Thank you to all the reviewers for their helpful feedback. We have uploaded our response to your comments, and we hope that we have satisfactorily responded to your concerns and suggestions. Thank you again - your comments have made our work better.

---

## [Decision Letter · Decision Letter 1]

7 Dec 2025

Dear Dr. Keenan,

Thank you for submitting your manuscript to PLOS ONE. After careful consideration, we feel that it has merit but does not fully meet PLOS ONE’s publication criteria as it currently stands. Therefore, we invite you to submit a revised version of the manuscript that addresses the points raised during the review process.

We look forward to receiving your revised manuscript.

Kind regards,

Jafar Kolahi

Academic Editor

PLOS One

Journal Requirements:

Additional Editor Comments:

Change the title to "Man vs. Machine: Multi-Country experimental evidence on the quality and perceptions of AI-generated research blog content"

Reviewers' comments:

Reviewer's Responses to Questions

**Comments to the Author**

Reviewer #1: All comments have been addressed

Reviewer #2: All comments have been addressed

Reviewer #3: All comments have been addressed

2. Is the manuscript technically sound, and do the data support the conclusions?

Reviewer #1: Partly

Reviewer #2: Yes

Reviewer #3: Yes

3. Has the statistical analysis been performed appropriately and rigorously?

Reviewer #1: Yes

Reviewer #2: Yes

Reviewer #3: (No Response)

4. Have the authors made all data underlying the findings in their manuscript fully available?

Reviewer #1: Yes

Reviewer #2: Yes

Reviewer #3: Yes

5. Is the manuscript presented in an intelligible fashion and written in standard English?

Reviewer #1: Yes

Reviewer #2: Yes

Reviewer #3: Yes

Reviewer #1: General Comments

The authors have substantially improved the manuscript in this revision. They have adequately addressed the major concerns regarding sample generalizability and the interpretation of "accessibility" within a highly educated demographic. The introduction of a dedicated "Discussion" section significantly strengthens the paper by placing the findings in the correct context and acknowledging the limitations of the Flesch-Kincaid metric for non-native English speakers. The statistical approach (OLS) has been justified satisfactorily.

Major Weaknesses: There are no remaining major weaknesses. The authors have successfully reframed the study to reflect that the results apply to a specific, policy-relevant, highly educated audience rather than the general public.

Minor Weaknesses: The primary remaining issues are editorial and formatting in nature.

Citation Formatting: There are numerous instances where internal reference codes (e.g., "Noy2023 [3]") remain in the text alongside the bracketed numbers. These must be cleaned.

Sentence Structure: Several sentences in the added text (Abstract and Discussion) are slightly convoluted and require syntactic polishing for clarity.

Typos: There are minor typographical errors introduced during the revision process.

Specific Comments

Abstract

Page 1, Line 119-120: The sentence "The sample in this study, while being the policy relevant sample who would engage with such policy-oriented academic research..." is wordy.

Edit: Simplify to: "The study sample consists of policy-relevant stakeholders who typically engage with academic research; they are highly educated and include thematic specialists."

Page 1, Line 122: "findings indicate that their view of 'accessibility' of research is such that..."

Edit: Clarify to: "findings indicate that this audience interprets 'accessibility' differently, preferring..."

Introduction

Page 4, Line 6: The citation format "Arnautu2021 [1]" contains the author/year key.

Edit: Ensure all in-text citations follow the journal style (e.g., just "[1]" or "Arnautu and Dagenais [1]"). This error repeats throughout the manuscript (e.g., Line 9 "Korinek2023 [2]", Line 13 "Noy2023 [3]").

Page 4, Line 14: "Jiao2023" appears without brackets.

Page 4, Line 15: The sentence starting "In addition, some studies highlight..." is a good addition regarding non-native speakers, but ensure the citation "[5]" is placed correctly.

Page 4, Line 28: The phrasing "The seminal paper by [16] and a more recent paper... by [17]" is grammatically awkward.

Edit: Change to "The seminal paper by Akerlof [16] and a more recent paper by Gans [17]..."

Methods

Page 13, Line 406: "While developed for U.S. school grades a limitation is that it was developed for American English..."

Edit: This sentence is grammatically broken due to the insertion of the limitation clause. Rephrase: "A limitation of this metric is that it was developed for American English and U.S. school grades; thus, it may contain measurement error when applied to this context."

Results

Page 17, Line 487: The phrase "using the adjusted p-values" is used.

Clarification: Ensure consistency in terminology between "adjusted p-values" and "RI p-values" established in the methods.

Page 22, Line 544: "This result could point to a behavioural bias against AI as this unfamiliarity increases ambiguity."

Comment: This is a logical inference and well-placed.

Page 23, Line 568: The sentence "We conduct a pre-specified analysis given some evidence that gender influences perceptions about AI Aldasoro2024 [59] with privacy..." contains a formatting error with the citation key and runs on.

Edit: Remove "Aldasoro2024" and fix punctuation.

Discussion

Page 26, Line 663: "In fact, the demographics of our sample... is likely the most important driver of our result."

Comment: This is a crucial acknowledgment that strengthens the paper's validity.

Page 26, Line 667: "An additional limitation is that our measure of 'readability'. We note that..."

Edit: Fragmented sentence. Change to: "An additional limitation is our measure of 'readability'."

Page 26, Line 670: "...rendering them over- or under-stated."

Edit: "stated" should likely be "estimated" or keep "stated" if referring to the reporting itself, but "under-reported" is more standard.

Reviewer #2: The authors have addressed the comments fully. I recommend the publication of this timely, pressing article.

Reviewer #3: I read the revised manuscript with great interest and I feel that my initial comments received satisfactory solutions. I gained readability and fluence, reads well and responds to debates in the scientific community. However, there are minor issues that should be addressed.

1. In the current form, both in the Discussions section and in Conclusions, the authors present the idea that using AI in research is welcomed, desired, having merit etc. especially for academic blogging. Using a more careful language here is a must: depends on the culture or expectations of audiences and on other factors. While readers/audiences are more receptive (or less critical, depends on the point of view) when AI use is disclosed, there are sufficient examples when 100% human authored content is considered of premium value. Assessing that AI use is desirable seems more like a technophilic wishful thinking, rather than an established truth.

2. Following my comment above, I wonder why authors left the idea of ethical challenges hanging. The debate around disclosing AI use and using it at all has strong roots in research ethics, as shown by the authors in the literature review and in referring to ethical approval for their survey. However later in the Discussions or Conclusions this idea is no longer present.

3. Minor linguistic verification of the manuscript is necessary. There are some sentences which need clarification. For instance "An additional limitation is that our measure of “readability”." This sentence is unfinished. Or, earlier, authors state that "This study has several strengths", but only two are highlighted, which cannot be considered several.

**Do you want your identity to be public for this peer review?** For information about this choice, including consent withdrawal, please see our Privacy Policy

Reviewer #1: **Yes:** Wissem Dhahbi

Reviewer #2: No

Reviewer #3: **Yes:** Assoc. Prof. Mariana Cernicova-Buca

---

## [Author Response · Author response to Decision Letter 2]

19 Jan 2026

Thank you for all the helpful comments, which have made the manuscript stronger. We have addressed all the comments and kindly ask you to refer to your response to reviewers document for more details.

---

## [Editor Report · Decision Letter 2]

29 Jan 2026

Man vs. machine: Multi-Country experimental evidence on the quality and perceptions of AI-generated research blog content

PONE-D-25-47038R2

Dear Dr. Keenan,

We’re pleased to inform you that your manuscript has been judged scientifically suitable for publication and will be formally accepted for publication once it meets all outstanding technical requirements.

Kind regards,

Jafar Kolahi

Academic Editor

PLOS One

---

## [Editor Report · Acceptance letter]

PONE-D-25-47038R2

PLOS One

Dear Dr. Keenan,

I'm pleased to inform you that your manuscript has been deemed suitable for publication in PLOS One. Congratulations! Your manuscript is now being handed over to our production team.

Kind regards,

on behalf of

Dr. Jafar Kolahi

Academic Editor

PLOS One